# Distinguishing the Effects of Water Volumes versus Stocking Densities on the Skeletal Quality during the Pre-Ongrowing Phase of Gilthead Seabream (*Sparus aurata*)

**DOI:** 10.3390/ani13040557

**Published:** 2023-02-05

**Authors:** Zachary Dellacqua, Claudia Di Biagio, Corrado Costa, Pedro Pousão-Ferreira, Laura Ribeiro, Marisa Barata, Paulo J. Gavaia, Francesco Mattei, Andrea Fabris, Marisol Izquierdo, Clara Boglione

**Affiliations:** 1Department of Biology, University of Rome ‘Tor Vergata’, 00133 Rome, Italy; 2Ecoaqua Institute, University of Las Palmas de Gran Canaria, 35214 Telde, Gran Canaria, Spain; 3Laboratory of Evolutionary Developmental Biology, University of Ghent, 9000 Ghent, Belgium; 4CREA—Consiglio per la Ricerca in Agricoltura e L’analisi Dell’economia Agraria (CREA)—Centro di Ricerca Ingegneria e Trasformazioni Agroalimentari, 00015 Rome, Italy; 5IPMA—Instituto Portugues do Mar e Atmosfera—Research Station, 8700-305 Olhão, Portugal; 6CCMAR—Centre of Marine Sciences, University of the Algarve, 8005-139 Faro, Portugal; 7UMR 7093, Laboratoire d’Oceanographie de Villefranche (LOV), Sorbonne University, 06230 Villefranche-sur-Mer, France; 8Associazione Piscicoltori Italiani, 37135 Verona, Italy

**Keywords:** morphometric quality, *Sparus aurata*, skeletal anomalies, stocking density, swimming space, tank volume

## Abstract

**Simple Summary:**

The development of skeletal anomalies in the early life stage of gilthead seabream (*Sparus aurata*) poses a significant challenge for farmers, affecting their profit margins, animal welfare, and the consumers’ perception of the aquaculture industry. Although many factors have been considered to be causative in the development of skeletal anomalies in marine finfish, the stocking density and available swimming space represent two key parameters which can be easily manipulated by the farmers during the critical phase of pre-ongrowing (prior to being placed in sea cages). This work aims at distinguishing which among tank volume and stocking density is the driving factor eliciting the development of skeletal anomalies during the pre-ongrowing phase in gilthead seabream, a productive cycle in which many skeletal anomalies can arise, particularly those affecting the vertebral axis. The results from this work indicated that particular cranial and axis deformities affected fish in higher incidences when they were reared at higher densities. The results are discussed through an eco-evo-devo approach in relation to the potential mechanisms at play affecting the increased prevalence of skeletal anomalies found. This research represents an intriguing contribution to aquaculture with results that can be applied directly to the production methods used by fish farmers to ameliorate the skeletal and morphological quality of farmed gilthead seabream.

**Abstract:**

Gilthead seabream (*Sparus aurata*) production is a highly valued aquaculture industry in Europe. The presence of skeletal deformities in farmed gilthead seabream represents a major bottleneck for the industry leading to economic losses, negative impacts on the consumers’ perception of aquaculture, and animal welfare issues for the fish. Although past work has primarily focused on the hatchery phase to reduce the incidence of skeletal anomalies, this work targets the successive pre-ongrowing phase in which more severe anomalies affecting the external shape often arise. This work aimed to test the effects of: (i) larger and smaller tank volumes, stocked at the same density; and (ii) higher and lower stocking densities maintained in the same water volume, on the skeleton of gilthead seabream fingerlings reared for ~63 days at a pilot scale. Experimental rearing was conducted with gilthead seabream juveniles (~6.7 ± 2.5 g), which were selected as ‘non-deformed’ based on external inspection, stocked at three different densities (Low Density (LD): 5 kg/m^3^; Medium Density (MD): 10 kg/m^3^; High Density (HD): 20 kg/m^3^) in both 500 L and 1000 L tanks. Gilthead seabream were sampled for growth performance and radiographed to assess the skeletal elements at the beginning and end of the experimental trial. Results revealed that (i) LD fish were significantly longer than HD fish, although there were no differences in final weights, regardless of the water volume; (ii) an increase in the prevalence of seabream exhibiting cranial and vertebral axis anomalies was found to be associated with increased density. These results suggest that farmers can significantly reduce the presence of some cranial and axis anomalies affecting pre-ongrown gilthead seabream by reducing the stocking density.

## 1. Introduction

Gilthead seabream (*Sparus aurata*) production is one of the main aquaculture industries in the Mediterranean, producing 258,754 tons of fish in 2019 [1]. Modern ongrowing farms (farms in which reared fish are fattened up to reach the commercial size of 200–350 g) often employ intensive strategies with high energy demands, tailored feeds, and high stocking densities, demanding elevated water exchange rates and increased levels of dissolved oxygen [2].

Generally, weaned juveniles (5 g) are pre-ongrown in land-based tanks for 8–10 months up to reaching 50–60 g, prior to being transferred to sea cages for the final grow out. In recent years, there has been a great deal of uncertainty with regards to the economic profitability for many gilthead seabream farmers. Since production costs often outweigh the profits due to a rapid market expansion in the 1980s, with consequent oversupply, a lower market value has been established in the last two decades [3]. The presence of skeletal anomalies in gilthead seabream production significantly hinders profitability [4]. Considering that gilthead seabream hatcheries produce varying amounts of suboptimal quality juveniles (i.e., with externally detectable skeletal anomalies), ongrowing farms are usually willing to pay premium prices for high quality fry. Unfortunately, this extra cost is not always effective: firstly, certain deformities are undetectable in early life stages, only becoming evident later on [5]; secondly, skeletal anomalies can arise at varying points throughout the life cycle [6].

Several factors may augment the economic uncertainty for farmers. The presence of deformed fish in farm products provokes higher management expenditures (energy, oxygenation, workforce, feeding for rearing deformed fish that are destined to die or be thrown out prior to being marketed), lower market value for suboptimal products, and/or the need for post-harvest processing to other less-marketable forms (fillet or pet food) [7,8,9].

The rearing cycle of pre-ongrowing is typically studied as an integrated system coupled with the ongrowing phase for economic, growth performance, or qualitative studies. The pre-ongrowing phase is rarely investigated disjointedly from ongrowing, although such studies could offer unique insights and opportunities to improve rearing strategies during an extremely pertinent phase. In ongrowing farms, the primary production costs come from the feed (46%) and fry (14%) [10]. These data highlight the importance of pre-ongrowing fry quality as they represent a significant cost for producers. For example, the occurrence of dental prognathism (underbite), which was found to affect 29% of juvenile gilthead seabream, had increased to 57% in the same lot once they had reached market-size [11]. Consequently, the high feed cost mentioned for grow-out farms could be effectively compensated for by identifying rearing procedures capable of reducing the percentages of deformed fish, thereby increasing the final marketable products.

Taking into consideration that the pre-ongrowing phase is mainly carried out in land-based facilities, it represents the last opportunity for farmers to apply quality controls and cull out deformed fish before their transfer into sea cages. Furthermore, the pre-ongrowing phase enables a more effective sorting/quality guarantee than culling conducted during or at the end of the hatchery phase. In fact, while fish with severe cranial anomalies could easily be culled out at the end of larval rearing due to the precocity of cranium osteogenesis, vertebral axis deformities are rare (or lethal if due to precocious notochordal defects) at this stage, often arising in more advanced developmental stages. Currently, routine practices during pre-ongrowing entails regular culling to both reduce the stocking density and to eliminate deformed fish from the production cycle [12]. This strategy can inflict additional stress on the animals [13] and requires a great amount of technical labor and know-how by the technicians.

Pre-ongrowing in tanks is generally carried out in intensive conditions that require frequent tank-cleaning as well as high water exchange rates and oxygen saturation [2]. While the pros and cons of choosing intensive versus semi-intensive rearing are weighed out by the farmers based primarily on the space availability, farm dimensions, and willingness to wait for economic returns (i.e., the quantity and size of fish that can be produced in the shortest time while minimizing the production costs), other incentives are gaining in popularity and economic competitiveness in European aquaculture. In particular, improved animal welfare, reduction of feed, water consumption, labor and energetic costs, organic labeling, and added-value marketing (gained by labeling schemes for high-quality products) all represent modernized approaches to animal production that can offer farmers a competitive edge and niche-market opportunities [14]. Obviously, the principal goal of producing both a high quantity and quality of fish remains paramount for the farmers. In order to efficiently optimize the desired quantity with the ideal quality, the best rearing parameters must be identified for each farmed species and life stage.

One particular biological issue that presents a significant loss in market value and overall profits is the presence of skeletal deformities in reared fish. In fact, in the European Union economic losses due to skeletal anomalies in fish farming were estimated to be around 50 million euros annually [4]. This value is likely the same or higher nowadays, as the problem has persisted [15] and oversupply/low demand in recent years is forcing farmers to optimize internal production efficiency to improve their profit margins [3]. The development of skeletal anomalies is undeniably complex due to the many etiologies and synergistic factors eliciting the development of skeletal anomalies [15,16,17]. Presently, many studies have been conducted to improve skeletal quality and reduce the presence of skeletal anomalies in gilthead seabream by using tailored rearing strategies to target life-stage specific requirements. Previous works among the main branches of research have focused their efforts on investigating the influence of dietary/nutritional factors [18,19,20,21,22,23,24], genetic components [25,26,27], and environmental dynamics [28,29,30,31,32] on the development of anomalies.

However, these strategies are not always applied to the actual production due to various reasons, such as expensive live or inert feed particularities (i.e., floatability, detectability, digestibility, etc.), tank/cage shape and water flow diversity, and novel environmental or technical challenges.

Nevertheless, two environmental parameters can be easily adapted and optimized by gilthead seabream farmers: the stocking density and the available swimming space. Previous studies have looked at stress parameter responses of different rearing densities in gilthead seabream [33,34,35,36,37,38], and others have compared the effects on skeletal anomalies of semi-intensive (‘Large Volumes’ and ‘Mesocosms’) to intensive rearing strategies [39,40,41]. Two recent studies on model fish species have also revealed that higher stocking densities provoke an increase in the presence of skeletal anomalies [42,43]. In a previous experiment carried out by Dellacqua and colleagues [44], it was found that reducing the stocking density during the hatchery phase improved the survival, increased final juvenile size, and reduced the presence of skeletal anomalies, while the increased water volume (swimming space) had a secondary positive effect by reducing the presence of jaw anomalies.

In this scenario, this study represents the first study aimed at delineating which, between tank volume and stocking density, is the main driver in the formation and proliferation of skeletal anomalies during the pre-ongrowing phase of gilthead seabream. The experimentation was designed to mimic industrial rearing management in order to be directly applied at a commercial scale. The expected results could greatly contribute to the optimization of gilthead seabream farming within the scenario of improving sustainability, profitability, and animal welfare. In order to facilitate the implementation of improved rearing strategies presented in this work, this study was conducted (i) at a pilot scale, following the standardized management practices typically applied during the pre-ongrowing phase; (ii) by testing three stocking densities (5, 10, and 20 kg/m^3^) suggested by the Italian Fish Farmers Association (API) and (iii) two tank volumes (500 L and 1000 L), which would typically be available for on-site use in any commercial gilthead seabream farm.

## 2. Materials and Methods

Experimental rearing was conducted at the Portuguese Institute of the Sea and Atmosphere (IPMA) facilities in Olhão (Portugal) during the late summertime for 63 days from August to October using gilthead seabream produced at the facility. Animal handling and experimentation was directed by trained scientists (following FELASA category C recommendations) and conducted according to the European guidelines on protection of animals used for scientific purposes (Directive 2010/63/UE of European Parliament and of the European Union Council), and the related guidelines and Portuguese legislation (Decreto- Lei 113/2013) for animal experimentation and welfare. IPMA’s Aquaculture Research Station was certified by the Direção Geral Direcção de Veterinária to conduct animal experimentation under the authorization 2018/12/17—025516.

### 2.1. Rearing Setup

Gilthead seabream juveniles (average weight of 6.7 ± 2.5 g and length of 7.8 ± 1.1 cm), obtained from different spawns and parents, were previously selected for skeletal anomalies based on an external inspection carried out by experienced technicians. The ‘non-deformed’ fish (hereafter referred to as T_0_) were stocked at 3 different densities LD: 5 kg/L; MD: 10 kg/L; HD: 20 kg/L in both 500 L and 1000 L water volumes, with a replacement rate of 100% tank water volume/hour, using sand-filtered natural seawater. The tanks for the 6 different experimental conditions were labeled LD1000, MD1000, HD1000, LD500, MD500, and HD500. The cylindrical tanks were all of the same diameter (1 m); however, they had different heights to hold two different water volumes (1000 L vs. 500 L). The stocking densities were periodically controlled during the experiment by sampling and weighing fish from each tank and removing excess fish to maintain the original biomass/L for each of the respective densities in the lots (Appendix A). The light regimen was based on natural light and photoperiod in the summertime in Olhão. The water temperature in the tanks ranged from 19–29 °C throughout the ~63 days of rearing (Appendix A). Oxygen diffusers were placed at the bottom of the tanks in order to maintain an O_2_ saturation level above 70% and DO > 5 mg/L, which was checked several times daily (Appendix A). Fish were fed 3% of their body weight/day with AquaGold 5 pellets (AquaSoja, Ovar, Portugal).

### 2.2. Sampling and Analyses

A sample of 161 T_0_ specimens were euthanized with 700 ppm of 2-phenoxyethanol and fixed in 1.5% PFA and 1.5% glutaraldehyde in 0.1 M sodium cacodylate buffer (pH 7.4), then transferred to 70% ethanol and successively radiographed (Gilardoni CPX 160/4 System, Unleaded film Kodak Mx 125, 55 Kv, 4 mA) for anatomical analyses.

During the trial, a total of 4769 individuals were sampled along 3 different time points from all of the experimental lots and weighed (wet weight—WW, weighed immediately once fish were anesthetized with 200 ppm of 2-phenoxyethanol) to calculate the growth in biomass and subsequently the density. After ~63 days of experimental rearing (T_F_), a total of 853 fish were euthanized following the same protocol that was applied to the samples from T_0_. T_F_ fish were measured for fork length, total length (TL), and wet weight (WW) (Appendix A). The Fulton’s condition factor (K) was determined on T_F_ samples based on the formula: K= 100 × W_t_/L^3, in which W_t_ represents the individual’s WW, while L represents the TL.

The Food Conversion Ratio (FCR), Specific Growth Rate (SGR), and % Weight Gain were calculated using the T_0_ and T_F_ weights according to formulas described in [45].

Specimens were radiographed with a digital DXS 4000 Pro X-ray (Carestream) (0.7 mm double top filter, 17 mm focal plane, with a single acquisition 120 s exposure time), and digitally visualized with the Carestream package at the University of Algarve. The anatomical investigation was carried out on the radiographs of T_0_ and T_F_ samples.

Dissections of the vertebral column were made on a total of 12 lordotic and 12 normal T_F_ fish individuated from the digital radiograph images, to assess potential differences in the calcium–phosphorous ratio between lordotic and non-lordotic vertebral centra (3 vertebral centra/fish). After removing attached soft tissues, the vertebrae were digested in nitric acid, and then the calcium and phosphorous mineral content was quantified using Microwave Plasma-Atomic Emission Spectroscopy (MP-AES; Agilent) at the University of Algarve.

Another 72 fish were also selected for specific anomaly types to be used for histology and sections of both deformed and non-deformed bones were dissected and were fixed as described above. Samples were then decalcified for 4 weeks by submersion in 10% EDTA and 0.5% PFA at 4 °C before being embedded in a GMA (glycol methacrylate) resin, according to the protocols described by Witten et al. [46]. Sections (5 µm) were cut on a standard rotary microtome (Microm HM360, Marshall Scientific (Hampton, NH, USA)) and stained with toluidine blue and Verhoeff’s elastin stain.

Datasets from each experimental group were first tested for the normality of the distribution (Shapiro–Wilk, Anderson–Darling, Lilliefors L, and Jarque–Bera JB tests) and then subsequently tested either with a parametric ANOVA or a nonparametric Kruskal–Wallis test for equal medians and a Dunn’s post hoc with Bonferroni correction.

### 2.3. Skeletal Anomalies Survey

Meristic counts and skeletal anomaly data (types and frequencies) were obtained from all of the 853 gilthead seabream individuals (T_F_) and 161 individuals (T_0_) by detailed examination of the radiographic images using FIJI [47]. Data was recorded using an alphanumeric code expressing the body region and the anomaly type (Appendix A). The examination was carried out based on the following assumptions: (1) incomplete fused bone elements were counted as discrete elements; (2) supernumerary bones with normal morphology were not considered to be anomalies but were included in meristic count variations; conversely, anomalously-shaped supernumerary elements were included among anomalies; (3) only clearly and unquestionably identifiable variations in shape were considered as skeletal anomalies: i.e., only the axis deviation associated to deformation of the vertebrae involved.

Skeletal anomaly data were then expressed in a raw matrix (RM) and used to calculate the frequencies (%) of each type of anomaly over the total number of anomalies, in each group (Appendix A). The RM was subsequently transformed into a binary matrix (BM), which was used to calculate the prevalence of individuals affected by each anomaly type (Appendix A). The following metrics were calculated for each group: (1) relative frequency (%) of individuals with at least one anomaly; (2) number of anomaly typologies observed; (3) average charge of anomalies (total amount of anomalies/number of malformed individuals); (4) relative frequency (%) of individuals with at least one anomaly; (5) frequency (%) of each anomaly type with respect to the total number of anomalies. The representative data from the RM and BM are expressed in tables and histograms. A Principal Component Analysis (PCA) was performed on the frequency of individuals affected by selected grouped anomaly types (Appendix A). The selection of the grouped anomaly types was based on the results obtained by testing the significance of differences in the frequencies of individuals affected by certain anomalies among the rearing conditions using a χ^2^ test and pairwise post hoc [48].

Axial deviations were also described by measuring the angle of bending from the apex, located at the point of greatest angular flexion with vectors drawn to either the dorsal or ventral (dorsal in the case of lordosis and ventral in the case of kyphosis) tip of the two vertebral centra preceding and the two successive centra following the apex angle (Figure 1). A modified categorization of angular class grouping made by Sfakianakis et al. [49] was applied to characterize 7 classes of axis deviations based on severity, according to the rankings shown in Table 1. A Kolmogorov–Smirnov (KS) test was applied to test for differences between the angular class frequency distributions among the lots.

Statistical analyses and graphs were performed using Past 4.02 [50] and Python 3.8.

### 2.4. Geometric Morphometrics

The total of 853 (T_F_) digitally radiographed gilthead seabream were analyzed with geometric morphometrics [51] and used to investigate differences in body shapes between the T_F_ individuals from different rearing conditions. Exactly 19 landmarks were manually selected on each digital radiograph, as displayed in Figure 2.

A Procrustes transformation [52] was applied to adjust landmark configurations for centroid size, eliminating potential effects that are irrelevant to shape. A two-dimensional between-group PCA (bgPCA), which is analogous to a Relative Warp Analysis [53], and a Linear Discriminant Analysis (LDA) were applied to the transformed x and y coordinates, calculating between-group variance to visualize the effects of the rearing condition on body shape. An Eigenvalue scale was applied to the results of the PCA in order to enhance visualization [54]. Thin Plate Spline (TPS) interpolation diagrams were obtained from the residuals of the Procrustes transformation [55] and visualized as spline deviations from the consensus along the weight matrix (the components of the bgPCA). The LDA differences between experimental groups were tested with a MANOVA (Wilks’ λ) [56]; the Mahalanobis squared distances were calculated for each pairwise group (i.e., LD1000 vs. LD500) and the differences between lots were tested for significance using a post hoc test with Bonferroni corrections.

## 3. Results

### 3.1. Performance Indicators

There were no sound differences in the survival, as mortality occurred with a total of 6 individuals (1 in LD1000, 3 in HD1000, 1 in MD500, and 1 in HD500).

There were no significant differences found in the final weight, FCR, SGR, nor % of Weight Gain (54.9 ± 1.9 g, 1.43 ± 0.20, 2.82 ± 0.07, and 7.16 ± 0.28%, respectively) among the tanks.

HD lots resulted in fish which were significantly (*p* < 0.05) smaller (TL) than the LD ones, regardless of the tank volumes (Figure 3a). Significant differences were also found in the Fulton’s condition factor (K) between the three densities tested in 1000 L, and between LD and the other two tested densities (MD and HD) in 500 L (Figure 3b). However, there were no significant differences in the final weight of fish between the conditions (Figure 3c).

### 3.2. Anatomical Analyses

Due to the disparity in the sharpness of the images between the film (T_0_) and digital (T_F_) radiographs, in this study only vertebral bodies and specific splanchnocranial anomalies were considered.

The observed anomalies affecting the vertebral region were deformities of vertebral centra (fusions, partial fusions, and hemivertebrae) and axis deviations. The anomalies observed in the head were opercular plate anomalies (Figure 4b) and anomalies affecting the maxillary, premaxillary, and dentary, defined as underbite (Figure 4c) or overbite (Figure 4d). The frequencies of individuals which resulted to be affected by each type of anomaly are shown in Figure 5.

The T_0_ lot resulted to display the highest anomalies’ charge: an average of three anomalies were detected in each deformed fish (Table 2). Elevated incidences of individuals with anomalies, particularly lordosis (38% of individuals; Figure 5) in the hemal region (32% of individuals), were observed in this lot. Furthermore, vertebral centra fusions affected a greater frequency (17%) of T_0_ than the T_F_ individuals (Figure 5).

Concerning the individuals in T_F_ lots, the majority had at least one anomaly, with occurrences ranging from 63 to 88% (Table 2). A clear density effect is discernable by the fact that both of the HD conditions displayed the highest percentages of individuals with at least one anomaly. However, for this metric, only HD500 resulted to be significantly different than the other conditions in a χ^2^ pairwise post hoc test (*p <* 0.001). A similar effect of density was recognizable in the number of observed anomaly types.

A secondary volume effect was also detectable: the 500 L lots displayed higher frequencies of deformed individuals than the 1000 L lots of equivalent densities even though these values were not significantly different (HD500 > HD1000, MD500 > MD1000, and LD500 > LD1000; Table 2).

The following sections report detailed descriptions of the observed anomalies in the different body regions.

### 3.3. Cephalic Skeleton

The T_F_ samples revealed higher frequencies of individuals exhibiting underbites, particularly in the HD500 lot (36.5%). The incidences of underbites among the conditions demonstrated a clear density effect as the frequency of affected individuals linearly increased with increasing density for both volume treatments (LD < MD < HD; Figure 6a). Additionally, a volume-effect was revealed by an augmented occurrence of affected fish in the 500 L volume conditions compared with their 1000 L equivalent density counterparts (LD1000 < LD500, MD1000 < MD500, HD1000 < HD500; Figure 6a). Although rarer overall, even overbites tended to be more frequent when the density was greater in the 500 L conditions (LD < MD < HD; Figure 6b), whereas the frequency in the LD1000 lot was less than those observed in the other two 1000 L lots (HD and MD), which displayed nearly equal frequencies (LD < MD ≅ HD; Figure 6b).

Conversely to underbites, the incidences of gilthead seabream with overbites did not display the same volume effect, as the frequencies varied haphazardly and the differences were relatively small (LD1000 < LD500, MD1000 > MD500, and HD1000 < HD500; Figure 6b). Nevertheless, the differences witnessed in the frequencies of individuals affected both by underbites and overbites did not result to be significant among the experimental lots (Pearson’s χ^2^ test: *p* values = 0.2 and 0.09, respectively).

In the case of opercular anomalies, which were inspected only from the left side of the fish, greater frequencies of fish affected by opercular anomalies were associated with greater densities and discernable in both volume groups (LD < MD < HD; Figure 6c). Analogous to what was found in the case of underbites, a secondary volume effect was also detectable as the smaller volume lots displayed higher incidences than the large volume density equivalents (LD1000 < LD500, MD1000 < MD500, and HD1000 < HD500; Figure 6c). Conversely to the jaw anomalies, these differences in frequencies of fish affected by opercular anomalies resulted to be significant between LD1000 vs. HD1000, LD1000 vs. MD500, LD1000 vs. HD500, and MD1000 vs. HD500 (Figure 6c).

### 3.4. Axial Skeleton

Vertebral body counts ranged from 22 to 25 in the LD1000 and MD1000 lots, while in the other lots the counts ranged from 22 to 24 (Appendix A). When analyzing increasing densities, a tendency towards lower vertebra counts appears (Appendix A) mainly due to higher occurrences of vertebral centra fusions, recognizable by the presence of a fully remodeled (amphicoelous) centrum and the presence of more than one neural or hemal arch. Specimens with vertebrae fusions and lower vertebral counts in the LD and MD varied between 0–16.7%, conversely to the HD lots displaying higher frequencies (28% and 52.6% for HD1000 and HD500, respectively). However, the overall frequency of individuals exhibiting fused vertebrae regardless of the vertebral body count did not vary greatly between the lots (8.6–11.7%). Although lower percentages were observed in the LD lots with respect to MD and HD ones (Figure 7a), these differences were not significant among the lots (χ^2^ test).

Concerning axis deformations, 42% of all individuals presented lordosis and 6% kyphosis. The histogram of the frequencies shown in Figure 7b displays a progressive increment of individuals with lordosis associated with increasing density. The highest frequency of lordotic individuals was found in HD1000 (52.3%) followed by HD500 (50.3%), while the other conditions displayed lower frequencies which were in the same order of magnitude for each density group, indicating a clear density effect, but no secondary volume effect (HD1000 (52.3%) > HD500 (50.3%) > MD1000 (37.4%) ≅ MD500 (37.4%) > LD1000 (31.4%) ≅ LD500 (32.3%); Figure 7b). Differences between the frequencies of LD and HD individuals affected by lordosis were statistically significant based on pairwise comparisons (Figure 7b).

The region most commonly affected by lordosis among all of the conditions was the hemal region with the HD1000 lot, followed by the H500 lot, displaying the greatest frequency of affected individuals (49.2% and 44.2%, respectively; Figure 8). Furthermore, the severity of lordosis, expressed by angular classes (Table 1), resulted to be significantly different among the experimental lots (Figure 9; Table 3), although no clear trend was discernable.

Kyphotic gilthead seabream were less frequent than the lordotic ones; however, an effect of density was detected between all of the 1000 L lots (LD < MD < HD), while in the 500 L conditions the trend is upheld only between HD and the other two conditions (LD500 (2.4%) ≅ MD500 (2.1%) < HD500: 12.2%; Figure 7c). Overall, the highest prevalence of individuals displaying kyphosis was found in the HD500 (12.2%) followed by the HD1000 (8.6%), and the χ^2^ pairwise post hoc test with Bonferroni corrections confirmed that significant differences were found between LD1000 vs. HD1000, LD1000 vs. HD500, HD500 vs. MD1000, HD500 vs. LD500, and HD500 vs. MD500 (Figure 7c). As far as the effect of water volume (1000 L vs. 500 L conditions with the same density) on the frequencies of kyphotic individuals is concerned, none of the differences observed resulted to be statistically significant (Figure 7c). Neither were there any significant differences in the distributions of severity classes (Appendix A), contrary to lordosis.

### 3.5. Histological Evaluations

Histological analyses highlighted interesting features of some vertebral anomalies. Partial (Figure 10) and complete vertebral body fusions display a pronounced dorsal–ventral asymmetry. Presumably, Figure 10a is representative of an early stage hemal fusion, in which the hemal arches begin to first fuse, with a narrowing of the ventral intervertebral space, which still persists (Figure 10b,c). Lateral bending in the orientation of the collagen fibers constituting the trabeculae was evident in the fusing centra in Figure 10b as well as in Figure 11b. In a complete fusion of hemal centra (Figure 11 and Figure 12c,d), the components of the intervertebral space (the collagen type-II based notochord sheath and its outer elastin layer) are still present ventrally, whereas modelling has already progressed dorsally (Figure 12c,d). In the case of hemal lordosis, the histological investigations reveal morphological changes in the shape of the centra, confirming the radiographic observations. All the components of the intervertebral spaces, neural, and hemal arches are maintained and healthy (Figure 13).

### 3.6. Principal Components Analysis

A PCA was applied to six anomaly types in order to enhance the visualization of differences between the conditions via an ordination plot. The anomalies considered were underbites, overbites, opercular anomalies, fusions, lordosis, and kyphosis (Figure 14). This PCA explained a total of 91.31% of the variance in the first two axes (76.62% and 14.69%, respectively for the first and second axes). In this ordination, a clear effect of density is distinguishable: the HD conditions were positioned towards the negative direction of component 1, in the same direction of the skeletal category vectors; MD conditions were positioned towards the center of the plane and LD groups plotted in the positive direction of component 1. Although component 2 explains less of the variance (14.69% vs. 76.62%), it hints at the secondary volume effect, detectable by the positioning of 500 L lots (green circles) in the negative hemiplane of component 2 (in proximity of the opercular and underbite anomaly vectors), in opposition to the 1000 L lots.

### 3.7. Geometric Morphometrics

A bgPCA was applied to the residuals from the Procrustes transformation derived from landmarks annotated on the radiograph images. In this PCA, the ordination model revealed that inter-group differences can be explained by 62.98% of the variance on the first axis and 24.01% of the variance on the second axis (Figure 15a). The LD lot centroids are both located on the positive plane of component 1, while the other lots are positioned on the opposite space in the negative hemi-plane of component 1. The component 2 enhances differences between the LD500 and LD1000 (quadrant 1 and 4, respectively) and between the HD (quadrant 2) and MD (quadrant 3) conditions (Figure 15a). Furthermore, lots with the same density and different volumes are positioned proximally to each other, suggesting a sound and strong effect of rearing density on fish shape. Thin plate spline (TPS) deformation grids, which were superimposed on the PCA in Figure 15a, assisted in further discerning shape differences among the conditions. The TPS deformation grid in the positive hemi-plane of component 1 (proximal to the LD lot centroids) shows that LD individuals exhibited a slenderer shape than the roundish shape of the MD and HD individuals located on the negative hemiplane of component 1 (Figure 15a). In particular, the differences reside mainly from landmark convergence within the caudal peduncle and divergence in the frontal-rostral region.

The LDA (Linear Discriminant Analysis) ordination explained 54.5% and 27.1% of the total variance in the first two axes, respectively (Figure 15b). Additionally, the LDA further enhanced the detection of shape differences between the groups displaying a strong effect of density as the LD condition centroids are positioned in the second and the third quadrants, in the negative hemi-plane of LD1, while the MD and HD centroids are located in the positive hemi-plane of LD1, in the first and the fourth quadrants, respectively. The LDA confirms that the centroids of the lots reared with the same density are located proximally to each other, independently from the tank volumes.

Lastly, a MANOVA (Wilks’ λ) applied to the uniform and non-uniform scores derived from the weight matrix revealed significant inter-group differences, while a pairwise test between the Mahalanobis squared distances validates differences between particular conditions (Table 4, Bonferroni corrected *p*-values, *p* < 0.00001). In particular, the greatest Mahalanobis squared distance was 14.9, which was found between LD500 and HD500.

### 3.8. Calcium-Phosphorous Mineral Content

Results from the mineral quantification revealed the absence of significant differences in the Ca/P ratios between malformed lordotic vertebral bodies and normal vertebral bodies, although a greater variability of Ca/P ratios from lordotic vertebral centra than normal centra is discernable (Figure 16).

## 4. Discussion

This study has demonstrated that pre-ongrowing gilthead seabream stocked in low densities (LD) for two months were significantly longer than those reared in higher densities, regardless of the water volume. The fact that individuals among the conditions were not heavier but presented significant differences in the condition factor K indicated that LD gilthead seabream acquired a slenderer profile than the fish reared in MD and HD, which appeared more roundish, as confirmed by the geometric morphometrics analyses. Both of these features could be due to the higher prevalence of lordotic fish found in high density conditions since the presence of axis deviations affecting the length and the body height could have given the deformed fish a stumpy body aspect. A previous study investigating growth and stress parameters on adult (commercial size: 217 ± 1.9 g) gilthead seabream reared for 6 weeks at 5, 10, and 20 kg/m^3^ found fish with greater weights in the LD conditions than in the HD conditions [33], confirming that the density can influence the size profile even in larger adults. The results from geometric morphometrics demonstrated that after only 63 days of rearing gilthead seabream sub-adults in the six different experimental conditions, significant differences in shape between the lots were already established. This seemed to be a response to the stocking density, even though the fish belonged to the same initial starting group of gilthead seabream from different origins. This is striking since morphometric analyses typically enhance shape differences between different populations and/or species from different ecological (i.e., feeding habits; ref. [56]), evolutive, or phylogenetic contexts [57,58], or in the same species, between differences in size classes [59], origins [60], life-stages, and/or long-term environmentally distinct factors (i.e., between wild and reared gilthead seabream; ref. [61]). It is likely that the differences we found after only 63 days of experimental rearing, in a relatively advanced life-stage dominated by isometric growth in gilthead seabream [62], were possibly due to the greater frequency of deformed individuals determined by HD conditions. The graphical representation in Appendix A, in which the spline of the centroid of a non-lordotic LD1000 individual is compared to the TPS of a lordotic HD500 individual, epitomizes this postulation.

The present study highlighted that when the effects of the rearing density were applied and analyzed separately from those of the tank volume, both may have exerted some effects on the skeletal phenotype in gilthead seabream. However, these effects varied according to the skeletal region. For example, while an effect of stocking density was evident in the frequency of opercular anomalies and axial deviations, no clear effects were identifiable regarding the incidence of vertebral fusions.

Overall, the most frequently observed anomaly found in all of the lots was lordosis. This anomaly is particularly problematic for gilthead seabream farms as its presence greatly reduces the products’ potential value. In this study, we found that the most common region of the body affected by lordosis was the hemal region. This finding is in accordance with other studies on the presence of hemal lordosis in gilthead seabream and has already been recognized to be frequently present in reared fishes [63]. Furthermore, the severity of lordosis can range from light deformations, which do not present any clear effects on the external morphology, to severe, with apparent impacts on external shape [64]. The increased severity of lordosis based on the angle of curvature was found to be significantly related with the number of affected vertebral centra in European seabass [49], therefore angular measurements can represent a reliable proxy for overall shape changes in the vertebral column. The classification of lordosis severity could also have useful applications for studies investigating recovery, due to recent findings that, in certain cases, affected fish can recover from lordosis [65,66,67]. In this current study, the classification of lordosis severity demonstrated that, at this life stage, gilthead seabream show a wide array of severity and that differences among the distributions of severity classes significantly differed in response to the density. This is the first study that has statistically confirmed differences between severity classes, laying the foundations for future applications and studies of axial deviation severity. The classification of different degrees of severity could also be useful for future applications on the recovery potential. For example, farmers could screen for slight axis deviations and apply different rearing strategies (such as reduced stocking density) to effectively recover those fish before the severity increases beyond a threshold of no-return, saving both time and money.

Lordosis had been initially proposed to be present only in association with improper inflation of the swim bladder [68,69,70]. However, Boglione et al. [71] found lordotic seabass juveniles with functional swim bladders, and Andrades et al. [72] described seabream larvae displaying a lordotic notochord even prior to the normal inflation of the swim bladder, suggesting that other etiologies may be responsible for eliciting this malformation. Some studies, in fact, have suggested that the primary causes for lordosis in reared fish are linked to the swimming activity [49,65,73,74] and that lordosis is an adapted response to increased mechanical loading exerted by trunk muscles [75]. This hypothesis seems to be corroborated by the healthy appearance of lordotic vertebrae from our histological analyses. Additionally, the finding of no significant differences in the calcium-phosphorous ratio (Ca:P) between lordotic centra and normal centra suggests that lordosis could be either triggered by other metabolic factors or by biomechanical responses. Similarly, to our findings, Boursiaki et al. [76] found no differences in the Ca:P ratio between scoliotic and non-scoliotic centra in seabream individuals. In our study, we observed a variability in the values of Ca:P ratio in the lordotic vertebrae. Previous studies have found that increased flow improved the growth rates in seabream [32,77]. Enhanced growth due to exercise can occur when fish are subjected to higher water replacement regimes, with the exercised fish experiencing a replacement rate of 700 L/h which establishes hyperplasia and hypertrophy in the white muscles [78,79]. Although the growth was improved in higher flow regimes in previous studies, a recent study by Palstra et al. [32] found an association with high flow regimes (~0.78 m/s) and increased frequencies of lordotic gilthead seabream. While in this current study an increased inlet flow velocity in the larger volume tanks was necessary to maintain the same water exchange rate as the small volumes, there were no differences in the frequencies of lordotic seabream between the different volume (flow) conditions. The water replacement rate and flow velocities tested in this current work, however, were lower than the replacement rate and inlet velocities from these previous studies (current study: maximum water exchange rate: 100%/h; flow velocities of ~0.73 and ~0.31 m/s for 1000 L and 500 L tanks at the inlets, respectively). Regardless, this potential effect of flow did not reveal itself in our experimental rearing since in cases of lordosis, no differences were present based on the different volumes (nor, consequently, flow velocity).

Contrary to water volumes, the stocking density did play a driving role in the prevalence of lordosis. The HD conditions could have influenced the fish behavior, based on the increased interactions among more individuals within the tanks (avoidance, escape, competition, aggression, hierarchies, etc.). Interestingly, in a previous work on the behavioral response of gilthead seabream to low stocking density, it was found that individual fish exhibited less overall movement within the tanks but a greater diversity in their responses to behavioral and social tests [80], while tighter interactions with consequent rapid changes of directions and burst swimming were more frequent in high density conditions. Burst swimming is characterized by high speeds maintained for a few seconds (<20 s) [81], rapid changes of direction by angular bending in the hemal region, and energy primarily supplied to myotomal white muscle through anaerobic processes [82]. All of these features yield potentially dire consequences for the (hemal) musculoskeletal system [34], with an onset of axis deviations in the hemal region induced by higher biomechanical pressure of muscle on underlying vertebrae.

Nevertheless, what has been discussed above should be considered in the context that the initial lot, T_0_, displayed a high presence of anomalies, whose occurrence in some cases were found to be reduced, and in others augmented in the final T_F_ seabream. The consequence is that the quality of the fish used to start the ongrowing phase define the quality threshold from which amelioration or worsening can be obtained by the predominant modulating factor: rearing density.

## 5. Conclusions

In the pre-ongrowing phase, the results of a primarily strong effect of density followed by a secondary benefit of increased water volume confirms previously found results regarding the incidences of skeletal anomalies in response to the rearing density during the hatchery phase [44]. Despite the fact that the life stage, production phase, genetic origin, and experimental location were different, some effects of the same physical driver (density) were identifiable in the same skeletal elements. Bear in mind that the differences in the life stage of fish under experimental trials were not negligible: the larval stage represents a critical and unique life phase, characterized by ontogenetic phases during which the differentiation of larval organs into the definitive (juvenile/adult) organs develop in succession, with consequences affecting the allometric/isometric growth, changes in habitat, trophic ecology, feeding modality, type of swimming, social responses, and behavior. The life stage under examination in the present study, the pre-ongrowing phase, is characterized by a more canalized [83] and exclusively isometric growth since differentiation has already been completed [62]. The genetic differences which characterize the gilthead seabream collected from different lots of different ages that only share the same weight-range are dramatic when compared to the same-batch origin of the specimens analyzed in the hatchery study.

The gilthead seabream skeleton seems to react to challenging environmental conditions by developing similar homeorhetic trajectories for modeling processes affecting the same bones. In particular, in both of the production/life phases, it was found that the use of higher densities determines higher occurrences of opercular, jaw, and vertebral axis anomalies. Additionally, it was found that the use of larger water volume mitigates the incidences of jaw anomalies, in both of the stages. However, considering that differences in occurrences of these anomalies were not significant, this study clearly individuates the stocking density as the main factor modulating the development of specific skeletal anomalies in gilthead seabream.

Future research carried out at different levels (biochemical, behavioral, and epigenetic) and on different species could actively help in unveiling the underlying mechanisms leading to the augmentation of anomalies in high density conditions.

## Figures and Tables

**Figure 1 animals-13-00557-f001:**
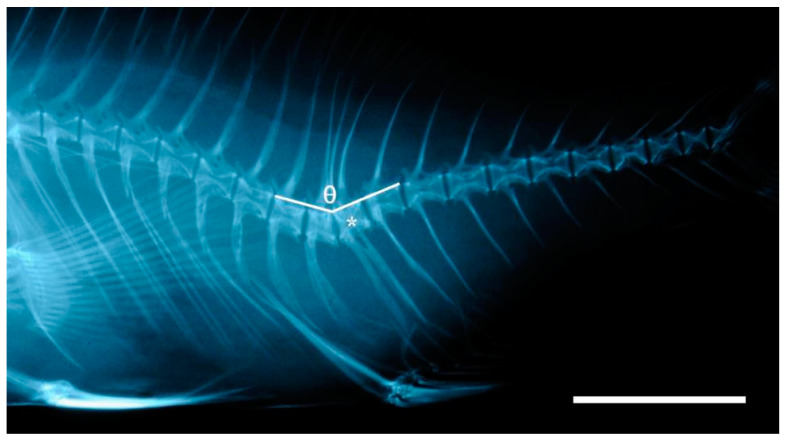
Gilthead seabream individual exhibiting a hemal lordosis (classified as ‘quite severe’ with an angle represented by θ of 142.5°) and a hemal vertebral body fusion marked by the asterisk (*). Bar = 1 cm.

**Figure 2 animals-13-00557-f002:**
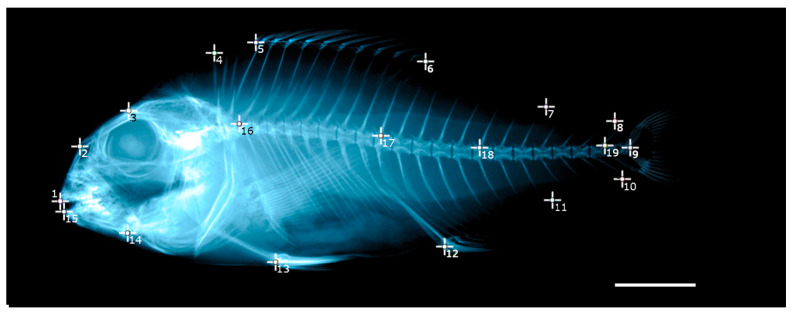
Radiograph of gilthead seabream individual with selected landmarks superimposed. Morphological features associated with each landmark: 1: Frontal tip of premaxillary; 2: rostral head point in line with the center of the eye; 3: dorsal head point in line with the center of the eye; 4: distal edge of the 1st (anterior-most) predorsal bone; 5: insertion point of the 1st dorsal hard ray; 6: insertion of the 1st dorsal soft ray pterygophore; 7: insertion of the last dorsal soft ray pterygophore; 8: dorsal concave inflection-point of caudal peduncle; 9: middle point between the bases of hypurals 2 and 3 (fork); 10: ventral concave inflection-point of caudal peduncle; 11: insertion of the last anal pterygophore; 12: insertion of the 1st anal ray pterygophore; 13: insertion of the pelvic fin on the body profile; 14: preopercle ventral insertion on body profile; 15: frontal tip of dentary; 16: neural arch insertion on the 1st abdominal vertebral body (carrying the first ribs and following the 2 cephalic vertebrae); 17: neural arch insertion on the 1st hemal vertebral body; 18: neural arch insertion on the 6th hemal vertebral body; 19: middle point between the pre- and post-zygapophysis of the 1st and 2nd caudal vertebral bodies. Scale bar = 1 cm.

**Figure 3 animals-13-00557-f003:**
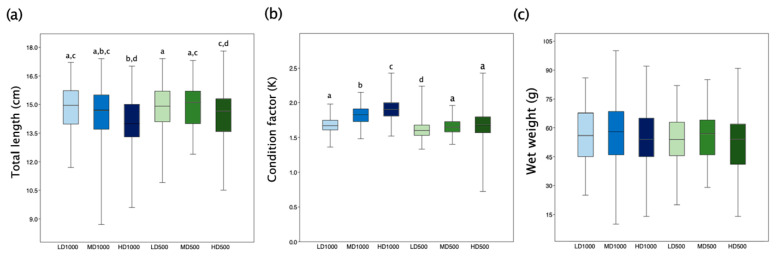
(**a**) Box and whisker plots of TL of T_F_ samples from the different experimental lots. (**b**) Differences in Fulton’s K factor between conditions. (**c**) Final wet weights among the lots. Box indicates the central percentile, the line inside the box indicates the median while the whiskers represent the minimum and maximum values. Different letters indicate significant differences (*p* < 0.05, Kruskal–Wallis, Dunn’s post hoc with Bonferroni corrections) among experimental lots.

**Figure 4 animals-13-00557-f004:**
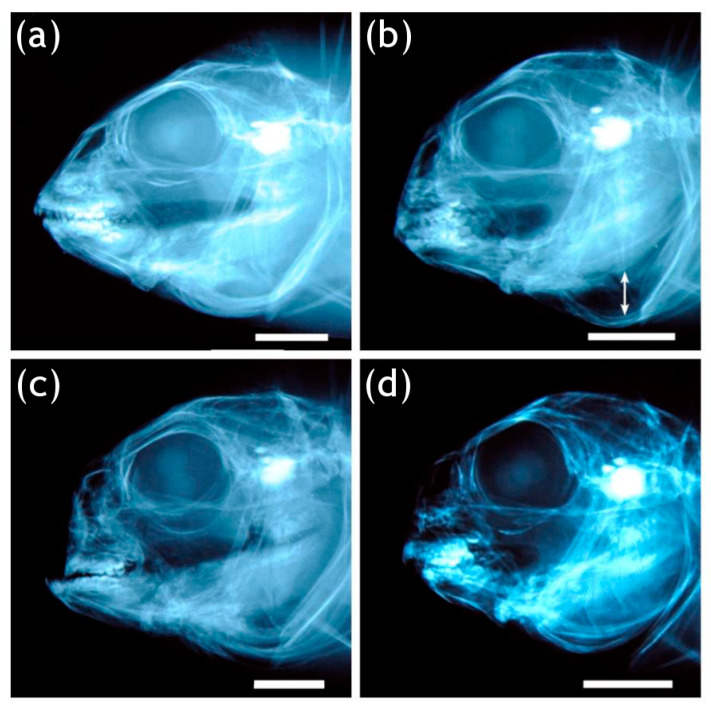
Splanchnocranium anomalies affecting gilthead seabream: (**a**) normal jaws and opercular plate; (**b**) reduced opercular plate, the arrow indicates a gap exposing the gill opening; (**c**) underbite: contemporary deformation of premaxillary and maxillary coupled with a protrusion of the dentary; (**d**) overbite: contemporary deformation (twisting, in this case) of the premaxillary and maxillary coupled with a reduction in length of the dentary resulting in an upper-jaw overlap. Bar = 5 mm.

**Figure 5 animals-13-00557-f005:**
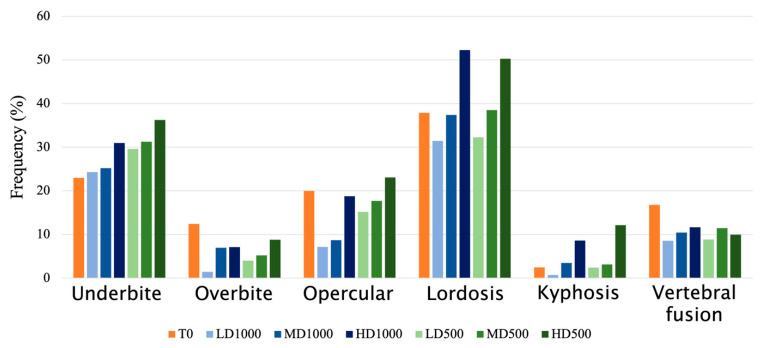
Frequency of individuals affected by the considered skeletal anomalies in the vertebral column and the cranium of gilthead seabream individuals evaluated from radiograph images in T_0_ and in each experimental density and volume condition (LD1000, MD1000, HD1000, LD500, MD500, HD500) at T_F_.

**Figure 6 animals-13-00557-f006:**
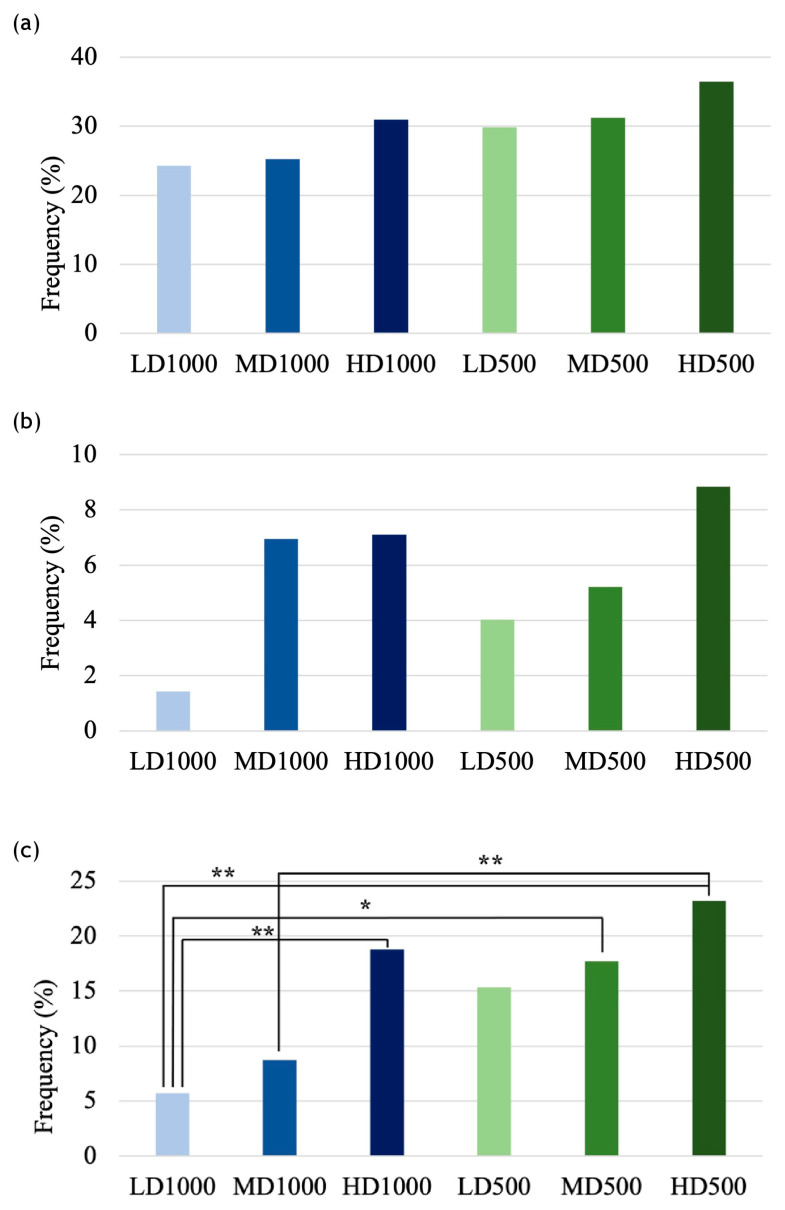
Frequencies of gilthead seabream individuals with (**a**) an underbite (%); (**b**) an overbite (%); and (**c**) a left-opercular plate anomaly (%), among the experimental conditions. Neither (**a**) nor (**b**) resulted to have any significant differences, while in (**c**) asterisks indicate pairwise significant differences (χ^2^ pairwise test with Bonferroni corrections, * = *p* < 0.05; ** = *p* < 0.005).

**Figure 7 animals-13-00557-f007:**
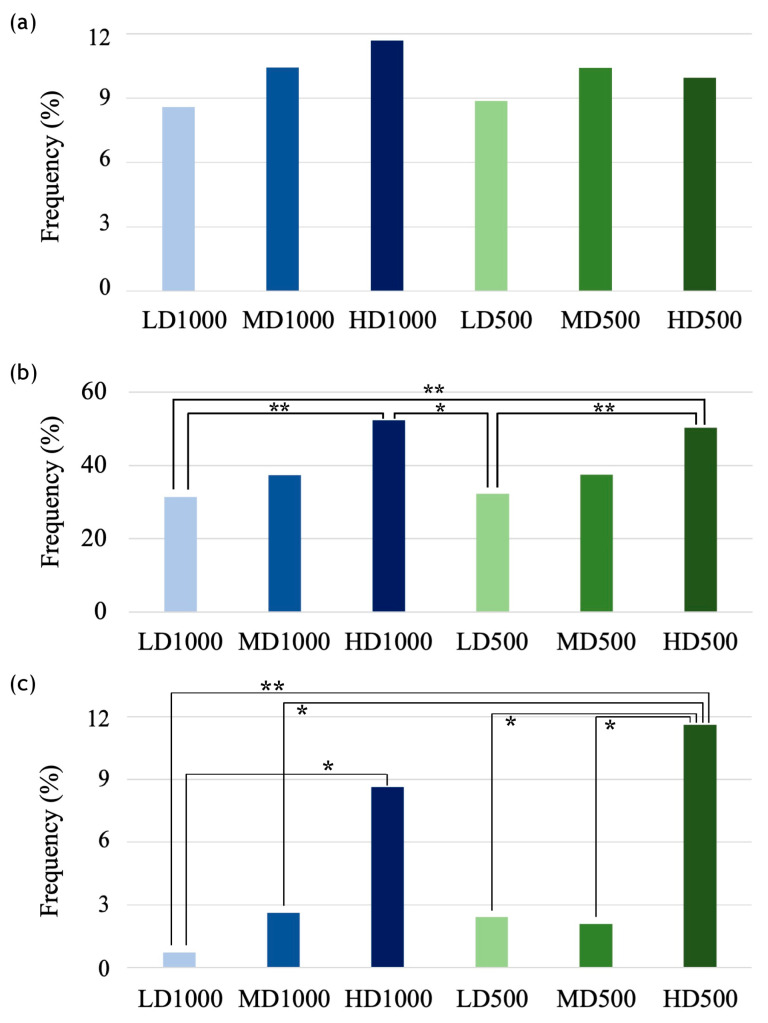
Frequencies of gilthead seabream individuals with (**a**) fusions of vertebral body centra (%); (**b**) lordosis in the vertebral axis (%); and (**c**) kyphosis in the vertebral axis (%), among the experimental conditions. (**a**) resulted to not have any significant differences, while in (**b**,**c**) asterisks indicate pairwise significant differences (χ^2^ pairwise test with Bonferroni corrections, * = *p* < 0.05; ** = *p* < 0.005).

**Figure 8 animals-13-00557-f008:**
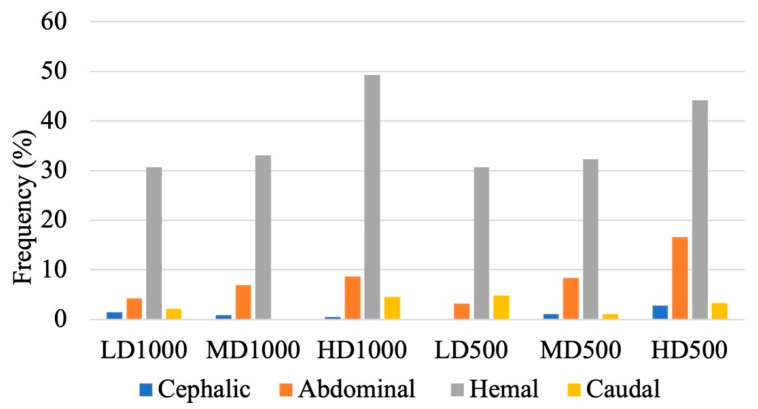
Subdivisions of vertebral regions in which individual gilthead seabream were affected by lordosis among the different experimental lots.

**Figure 9 animals-13-00557-f009:**
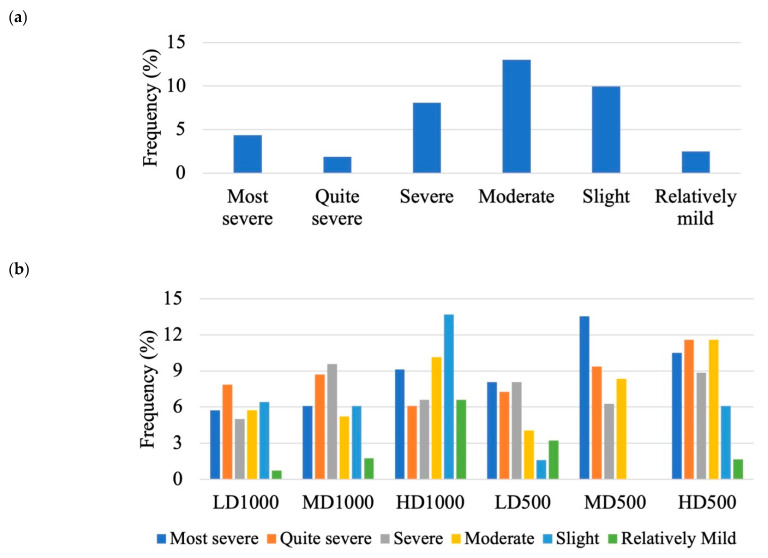
Distribution of different angular classes of lordotic curvatures based on the classes described in Table 1: (**a**) in the lot T_0_ at the beginning and (**b**) the different T_F_ lots of different densities and volumes at the end of the experimental rearing. Significant differences from the KS pairwise post hoc are described in Table 3.

**Figure 10 animals-13-00557-f010:**
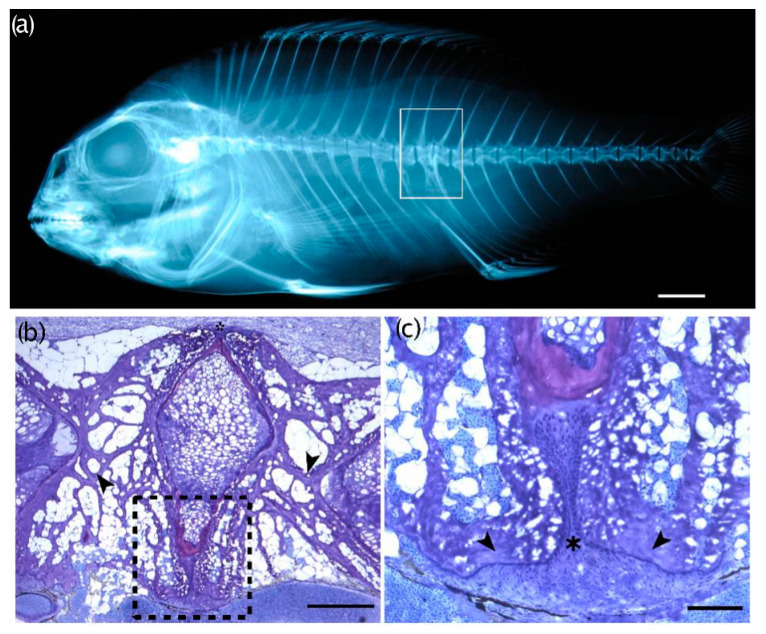
Compressed/partially fusing vertebral centra with an already fused dorsal zygopophysis. (**a**) Radiographic image of a T_0_ seabream exhibiting two compressed vertebral centra (rectangle). Bar = 5 mm; (**b**,**c**) Toluidine blue-stained sections of the deformed vertebral bodies shown in (**a**). The black arrowheads in (**b**) point to bending trabeculae associated with remodeling of the misshapen centra. Note the dorsal–ventral asymmetry, where the ventral regions of the two centra appear enlarged. Bar =1 mm; (**c**) higher magnification of the region indicated with the black dashed rectangle in (**b**): the asterisk marks the insertion point of the hemal arches in the ventral intravertebral space between the two centra. The black arrowheads point to the bone of the hemal arches. Bar = 0.1 mm.

**Figure 11 animals-13-00557-f011:**
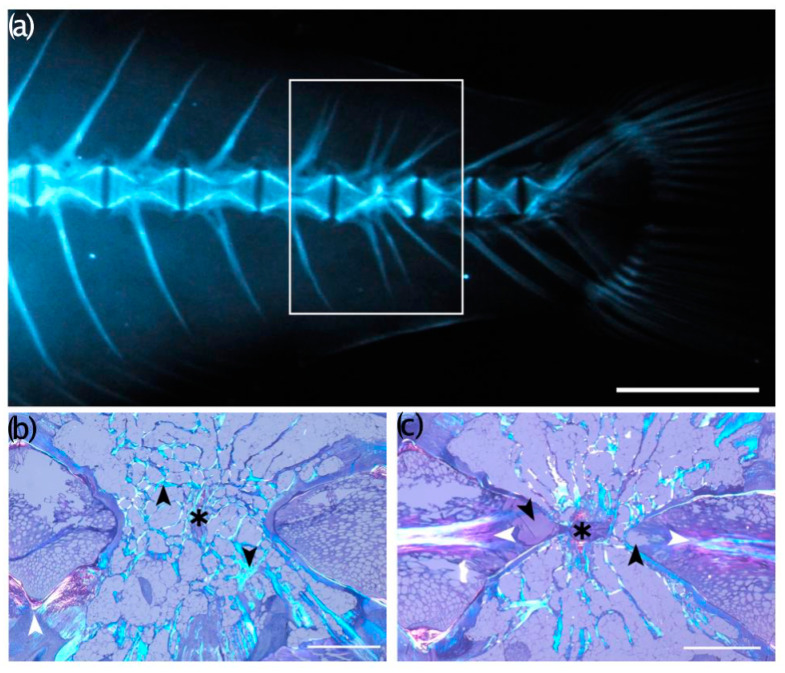
Complete fusion of vertebral bodies. (**a**) Radiograph image of a T_0_ seabream exhibiting a multi-vertebra fusion (rectangle). Bar = 5mm; (**b**,**c**) Toluidine blue-stained sagittal section of the multiple fusion event shown in (**a**), photographed with polarized light. The black arrowheads in (**b**) point to laterally bending trabeculae that surround the fusing vertebral centra. The presence of intravertebral space between the trabeculae of the fusing centra (asterisk) indicates a lateral deviation of the fusing vertebral bodies as in a localized scoliosis. The white arrowhead points to a normal vertebral body endplate. Bar = 1mm; (**c**) the black arrowheads point to putative reactive scar tissue of the notochord strand (white arrowheads), occurring proximal to the point of the fusing centra (asterisk). Note the cartilage (in pink) of the hemal and neural arches, located both dorsally and ventrally to the fusing centra above and below the asterisk in (**c**). Bar = 1mm.

**Figure 12 animals-13-00557-f012:**
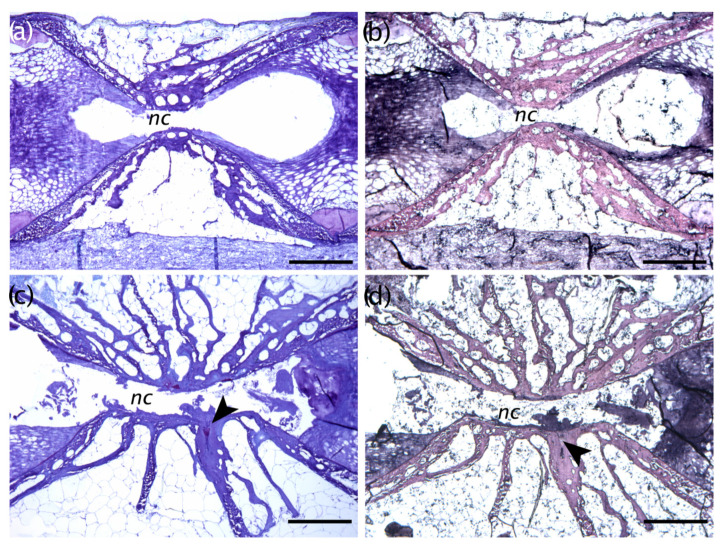
(**a**) Non-fused vertebral centra and, (**b**) multiple fusions among preural centra a, (**c**) Toluidine blue staining; (**b**,**d**) Verhoeff’s elastin-staining. The black arrowhead in (**c**) indicating that all of the components of the intravertebral ligaments are maintained; and in (**d**) the arrowhead indicates the persisting elastin layer in the fused intervertebral space. Bar = 0.5 mm.

**Figure 13 animals-13-00557-f013:**
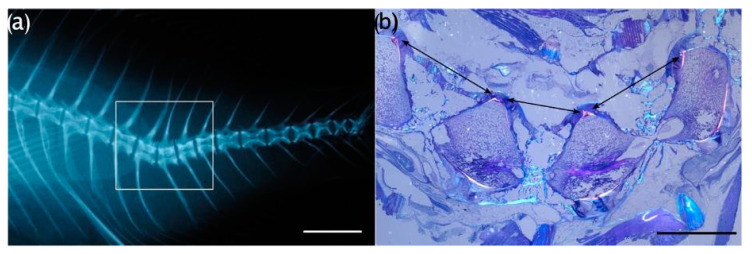
(**a**) Radiograph of a lordotic seabream (the white rectangle highlights the localized affected area). Bar = 5 mm; (**b**) Toluidine blue parasagittal section photographed with polarized light of the lordotic vertebral body centra shown in **a**. The double arrows highlight the different widths of the dorsal profile of lordotic vertebrae. Bar = 2 mm.

**Figure 14 animals-13-00557-f014:**
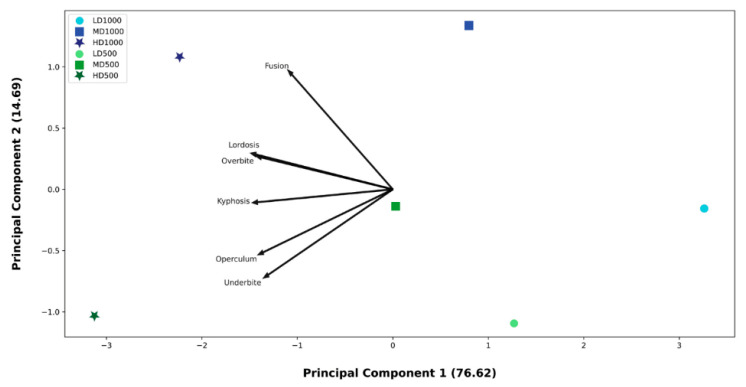
PCA ordination model applied to the frequencies of individuals affected by underbites, overbites, opercular anomalies, fusions, lordosis, and kyphosis in each experimental condition. The resulting ordination model indicates that the high-density conditions were highly associated with these skeletal anomalies.

**Figure 15 animals-13-00557-f015:**
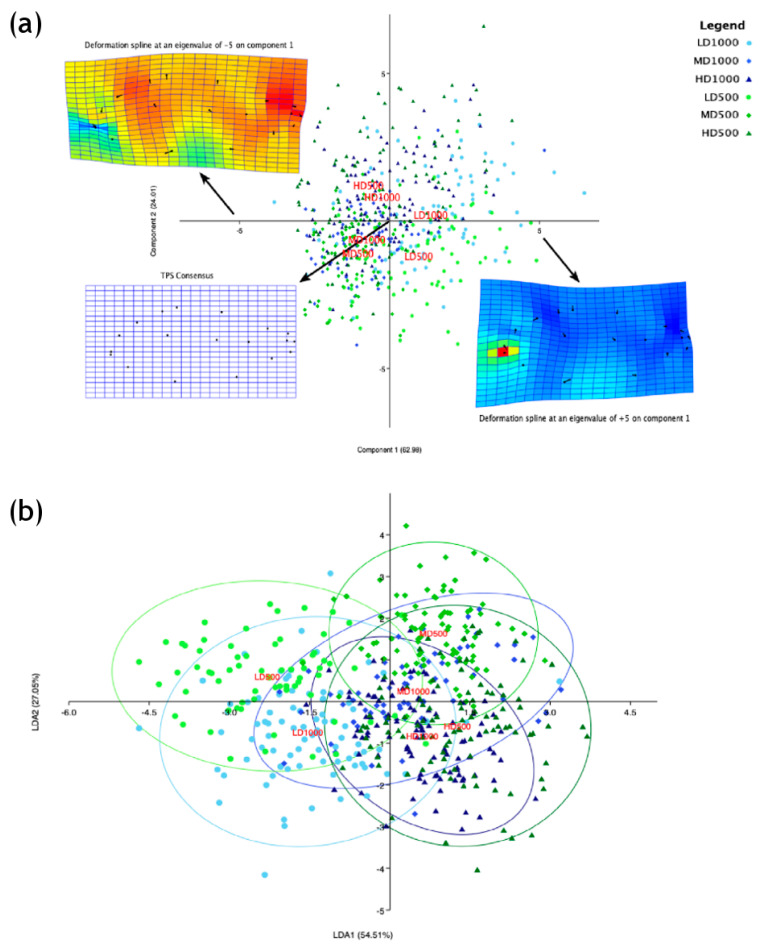
(**a**) bgPCA scatter with deformation grids plotted on the extremities of the first component. TPS deformations are color-coded in which cool colors (blue/green) represent convergence with respect to the consensus and warm colors (yellow, orange, and red) represent divergence. (**b**) Linear Discriminant Analysis (LDA) scatter results with 95% ellipticals and lot centroids, highlighting group shape differences among the T_F_ experimental lots of different densities and volumes.

**Figure 16 animals-13-00557-f016:**
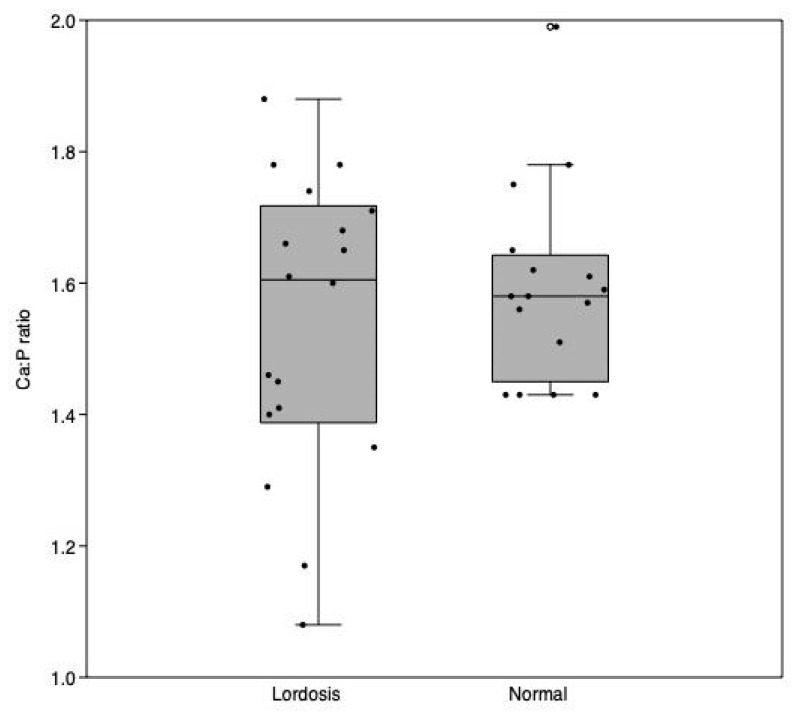
Calcium–phosphorous ratio in ppm (µg/mL) of Ca to ppm (µg/mL) P between lordotic and normal centra (three vertebral bodies). Each dot is indicative of a single individual while the central line denotes the median. (ANOVA, NS).

**Table 1 animals-13-00557-t001:** Ranking of different angular classes of axis deviation.

Most severe	<137.9°
Quite severe	138.0°–144.9°
Severe	145.0°–151.9°
Moderate	152.0°–158.9°
Slight	159.0°–165.9°
Relatively mild	166.0°–172.9°
Normal	173.0°–180°

**Table 2 animals-13-00557-t002:** Metrics calculated at T_0_ and in the experimental lots.

Lots	T_0_	LD1000	MD1000	HD1000	LD500	MD500	HD500
Number of observed specimens	161	140	115	197	124	96	181
% of individuals with at least one anomaly	79	63	69	80	69	75	88
Anomalies charge (n. of anomalies/affected fish)	3	2	2	2	2	2	2
Observed types of anomalies	21	18	23	23	17	21	23

**Table 3 animals-13-00557-t003:** *p*-values obtained from a Kolmogorov–Smirnov (KS) pairwise post hoc test with distribution of lordosis severity (excluding non-deformed individuals). Only significant (*p* < 0.05) comparisons are reported.

Pairwise Post Hoc	*p*-Values
LD1000 vs. HD1000	0.002
MD1000 vs. HD1000	0.002
HD1000 vs. LD500	0.002
HD1000 vs. MD500	0.02
LD500 vs. HD500	0.02

**Table 4 animals-13-00557-t004:** Mahalanobis squared distances inter-groups.

	MD1000	HD1000	LD500	MD500	HD500
LD1000	6.7184	6.5169	4.0499	11.3400	8.7823
MD1000		2.5247	9.1422	3.6365	3.9081
HD1000			11.321	6.9757	2.9881
LD500				11.575	14.898
MD500					5.905

Significantly different distances were found between all groups (Bonferroni correction, *p <* 0.0001).

## Data Availability

The data presented in this study are available in the Appendix A. Additional data presented in this study are available on request from the corresponding author. The data are not yet publicly available due to unfinalized database selection.

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
