# Peer review of "Distinguishing the Effects of Water Volumes versus Stocking Densities on the Skeletal Quality during the Pre-Ongrowing Phase of Gilthead Seabream (Sparus aurata)"

_animals, 2023, doi:10.3390/ani13040557_

Round 1

Reviewer 1 Report (Previous Reviewer 1)

The article captioned Distinguishing the effects of water volumes versus stocking densities on the skeletal quality during the preongrowing phase of Gilthead seabream (Sparus aurata) and MS id animals-2185709 is now revised in the light of he comments  that i have raised during first review. So now i recommend the article to be published in the journal. The quality of article has improved alot.

Author Response

Thank you for your useful and insightful review.

Reviewer 2 Report (Previous Reviewer 3)

The authors have seriously and appropriately addressed the criticisms I raised. In my opinion, the manuscript has improved a lot. I recommend publication in this journal. 

Author Response

Thank you for your insightful and well-thought reviews.

Reviewer 3 Report (Previous Reviewer 4)

As I wrote: "Although the hypothesis is interesting, the discussion about ROS does not contribute to a clear understanding of the influence of volume and density of fish in this research on the obtained results. Conversely, parameters directly related to ROS were not measured in this paper, so that ROS could be the basis for interpreting the obtained results." - this paragraph should be omitted because it gives the research scope that is not covered by the results

Author Response

Thank you for your well-thought and helpful review. We had originally tried to reduce speculation regarding ROS, however in this version the paragraph has been eliminated as requested. We believe this final version is satisfactory. 

This manuscript is a resubmission of an earlier submission. The following is a list of the peer review reports and author responses from that submission.

Round 1

Reviewer 1 Report

Due to flaws presentation, The MS require a significant improvement and i encourage is resubmission. 

Title: the word evaluation, effects are not cathy so change the tile little bit innovative.

Simple summary: as per the format of animals it is mandatory to put one simple summary section but is not there in MS. It indicates that authors are in hurry to submit half cooked article to a such reputed journal.

Abstract

Line 38-39, it has been mentioned that fishes were observed with lordosis, kyphosis, here my point of mention is that these abnormalities are due to deficiency of amino acids then how you correlated it with the stocking density and volume nod water whether fish are not getting sufficient amino acids in feed. What is the reason justification is required?

In addition, Scoliosis is not observed. It means fishes are getting pressure vertically only not horizontally. So, it needs an elaboration.

Introduction

The old references to be replaced with newer one. Preferably 80 % refences should be beyond 2017.

Line 1270130 need reorientation there should not be numerical digit only what the study been conducted should reflect.

Material and methods

Rearing setup

Experimental duration needs to be mentioned.

Line 177, is it specific growth rate or standard growth rate???????

If you have data for initial weight and length then you can calculate other indices also like FER, PER, Weight Gain, Weigh gain %, Average daily growth etc. and for that you can refer following references and must to include in this section.

1.      D.K. Meena, A.K. Sahoo, M. Jayant, N.P. Sahu, P.P. Srivastava, H.S. Swain, B.K. Behera, K. Satvik, B.K. Das, Bioconversion of Terminalia arjuna bark powder into a herbal feed for Labeo rohita: Can it be a sustainability paradigm for Green Fish production? Animal Feed Science and Technology, Volume 284, 2022.

2.      M. Jayant, M.A. Hassan, P.P. Srivastava, D.K. Meena, P. Kumar, A. Kumar, M.S. Wagde, Brewer’s spent grains (BSGs) as feedstuff for striped catfish, Pangasianodon hypophthalmus fingerlings: An approach to transform waste into wealth, Journal of Cleaner Production, Volume 199,2018, Pages 716-722.

3.      Geetanjali Yadav, Dharmendra Kumar Meena, Amiya Kumar Sahoo, Basanta Kumar Das, Ramkrishna Sen, Effective valorization of microalgal biomass for the production of nutritional fish-feed supplements, Journal of Cleaner Production, Volume 243, 2020, 118697, ISSN 0959-6526.

Results are ok.

Discussion need more refinement and coherence as per the results and based on previous research in similar fled and figure should be removing from discussion if it is essential it can be put in result section where ever applicable.

Conclusion: no text is there again its very ridicules that author even have not seen once the article before submission.

References: a through cross for reference need to be done for formatting.

Overall comments: the author tried to address the important aspect of culture system of seabream and that can be replicated in other fish culture also with slight modification. However, I am, very sorry to write and got upset how authors can be too causal in writing and submitting a manuscript in a such reputed journal. There are several mistakes and needs rephrasing, language need to be improved. So in view of above I am compel to reject the MS in its present form but encourage the resubmission in same journal.

Reviewer 2 Report

The authors investigated the effect of water volumes versus stocking densities on the skeletal quality during the preongrowing phase of Gilthead sea bream (Sparus aurata). They designed six treatments to test their hypothesis. This manuscript (MS) was clearly written and easy to understand. This work can help the sustainability of this species farming as few studies have been done on this topic. However, some major issues significantly compromised the quality of this MS.

Comments

·       Line 24, please delete “very”

·       Line 107, please say more clearly about the problems related to skeletal abnormality.

·       Line 112-115, is not relative to those studies. Please only focus on the topic of this MS.

·       Line 177-177, revise and make sure the formula and units are correct.

·       Line 265, why MANOVA??. You have six treatments and can do it two ways ANOVA with the main effect of density and volume. If the interaction was significant, you could unpack the original data. If not, you can compare the main effects.

·       Line 286, why Kruskal-Wallis??

·       Line 307, please explain more about T0. You did not mention that in the material and method; please explain there.

·       Line 360, again, is not clear why you use this method of statistical analysis.

·       Figure 7 and others, please clarify whether there was a significant difference or not.

·       Table 4, is not clear about the method. I think you should change and simplify the statistical analysis.

·       Please revise figure 10; some treatments are wrong and missed

·       Please move figure 16 to the supplementary file., it did not add anything new to the MS.

·       Please move figure 18 to the supplementary file., it did not add anything new to the MS.

·       In discussion, please try to focus only on important results and too much information will distract the readers from the most important points. Further, I suggest combining some figures and tables to visualise better for readers.

·       Here and elsewhere, report P uppercase and italic (P<0.05).

·       Throughout the MS, if there is no significant difference, no need to report P-value.

·       Please reorder the keywords alphabetically and capitalize each word.

·       Please write the abstract more numerically about the results. You can do it by adding their numbers in parentheses.

·       Here and throughout the MS, please first mention the common name plus scientific name, and for the rest of the MS, just report the common name.

·       Please update the introduction with recent works as many studies are available from the last two years, which were not included in this section.

·       Please mention the novelty of your work in the last paragraph of the introduction.

·        

Material and methods

·       Well-organized section. Clear fellow and all required details were provided.

·       Please mention how many percentages of water were exchanged each day if you have monitored.

·       For each analysis, please clarify how many fish were taken.

·       Some parts of the discussion are better updated with research in 2021 and 2022 as they refer to some old references. Please update the discussion with the latest studies as much as possible.

·       Although you wrote this section well, you can still improve it by answering these questions and annotating them to the discussion section. Why were these results observed? Discuss more possible reasons.

Tables and Figures

•            Please explain a little bit about your experimental treatments per each Table and Figure. Each Table and figure should represent enough information separately from the text.

•            Double-check the units and titles of all Tables.

•            Please mention in the footnote of all Tables which kind of statistical method you used for comparing the means.

Kind regards

Reviewer 3 Report

The manuscript written by Dellacqua and colleagues describes results of a study in which juvenile sea bream were reared under different water volumes and stocking densities. The analyses mainly involve effects on different parts of the skeleton. The rationale for the study is explained clearly and to good detail.

I do have concerns about the experimental design; namely, the lack of tank replications. The readout parameters were carefully chosen and thoroughly analysed, but difficult to interpret due to the lack of true replications. This is a major shortcoming in the experimental design. To test effects of a treatment, data were used from experiments whose treatments were not replicated (N=1; no tank replicates). This makes the tank the experimental unit, and fish the observational unit. I would add to that when analysing the data in Figures 6, 7, 8, etc., a two-way anova would be more appropriate. Data involve one quantitative dependent variable (frequency) and two categorical independent variables (tank volume and stocking density). 

While the data are of interest to farmers of this particular fish species, the insights derived from the work presented are not necessarily of interest to a wider readership. Mechanistic insights that come forward from the data are also limited. 

In general, a revision on writing style wouldn't hurt. Sentences are generally very long, with numerous clauses, parts of which are all too often in brackets.

 The discussion is elaborate. My suggestion would be to cut the number of words and bring the main message to the front. Also, avoid any speculation (e.g. lines 604-607; 670-672).

Minor points:

-       Line 125, which

-       Line145-146: Based on what considerations were the densities and water volumes chosen? How do they relate to sea bream culture practice, and/or to those used in the numerous studies that preceded the current study? Which of the experimental condition could be considered closest to "reference"?

-       Reasoning in Line 272-275 is contradictory

-       Although not affected, I still would suggest to include final body weight in fig.1 to visualize the variation in the data and between tanks.

-       Make an abbreviation list, e.g. in tabular form, to guide the reader through the many abbreviations used. Also reconsider carefully whether it is really necessary to make an abbreviation for almost everything (WW, TL, etc)

 The abstract needs thorough revision. It is difficult to read for several reasons. Some examples of why: 

-       an excessive use of adverbs (e.g. line 24 "very highly"); 

-       incorrect sentence structure (at lines 25-28); 

-       unnecessary details (line 32 where the experiment was conducted); 

-       vague terms ("performative data", "morphological quality") 

-       unexplained abbreviations (T0, TF, lines 36-37). 

-       Also, the beginning of the abstract lacks the rationale for the study (why pre-ongrowing phase? There evidence that this stage is of particular interest, as outlined in the introduction. Please provide it briefly in the abstract). 

-       At last, an implication of the study results is missing at the end of the abstract.

Reviewer 4 Report

Review comments for the Manuscript ID: animals-2102566

Effects of water volumes versus stocking densities on the skeletal quality during the preongrowing phase of Gilthead seabream (Sparus aurata)

The manuscript is attractive, well organized and contributes to a better understanding of the effect of stocking density and water volumes on rearing-induced skeletal anomalies of Gilthead sea-bream juveniles. Materials and Methods and Results clearly present the conducted research, but part of the Discussion is too ambitious. The manuscript contains novel features and deserve to be published after the consideration of the review comments recommended.

Material and Methods:

Lines 203-204: The term "discreet", I suppose, should be replaced with term "discrete".

Results:

Performance indicators -lines 271-287: It would be useful (if possible) to supplement the analysis of length and condition factor with a calculation related to specimens without skeletal anomalies and further strengthen the good discussion about the effect of deformations on the estimation of length and K . (lines 525-533)

Discussion:

Line 547: The statement/assumption that isometric growth dominated should be explained.

Lines: 574-577: The presented results do not clearly indicate that there was a recovery in some groups, especially since the hypothesis was not unambiguously statistically evaluated. It is necessary to provide an explanation that logically directs the obtained results to such a judgment.

Lines: 598-607: The obtained result related to the Ca:P ratio is useful for future research, but the ratio alone is not sufficient to suggest to downplay the metabolic etiology. It can be said that the results of the analysis of the Ca:P ratio do not indicate that the problem could be related to the metabolism of Ca and P.

Lines 627-643: The results of this research statistically confirmed the influence of fish density on lordosis, but the influence of sea bream behavior, which was not studied in this paper, was too ambitiously combined with the results of other studies that were not conducted under identical conditions. In addition, the frequency of different behavior of Gilthead sea-bream related to the rearing density has yet to be quantified and correlated with observations in skeletal anomalies. That's why I suggest a more prudent approach in connecting literature references with obtained results.

Lines 649-663: In the Materials and Methods (lines 158-161), the authors define the experimental conditions for dissolved oxygen in tanks by setting a minimum value: "an O2 saturation level above 70% and DO 159 >5 mg/L". I assume that this was supposed to eliminate the possibility of the oxygen concentration affecting the obtained results. That is why (?)different concentrations of dissolved oxygen were not correlated with the measurements in the presented Results. And then, the discussion tries to connect the assumed distribution of oxygen in the tanks (not measured), with the flow, with the behaviour and with possible hyperventilation of sea bream in an overly ambitious way, and the cited literature does not confirm but only helps in making the hypothesis. This can be a good basis for new research, but it goes beyond the scope of the actual measurements and achieved results. I suggest revising this part of the discussion within the framework of the obtained results and with the literature that is directly related to the discussed topic. In the case of overbite and opercular anomalies, there was a greater or lesser recovery (?), while in the case of underbite there was no recovery (?)….

Lines 664-672: Although the hypothesis is interesting, the discussion about ROS does not contribute to a clear understanding of the influence of volume and density of fish in this research on the obtained results. Conversely, parameters directly related to ROS were not measured in this paper, so that ROS could be the basis for interpreting the obtained results.